# A SELF-ORGANIZING MEMORY NETWORK

## ABSTRACT

Working memory requires information about external stimuli to be represented in the brain even after those stimuli go away. This information is encoded in the activities of neurons, and neural activities change over timescales of tens of milliseconds. Information in working memory, however, is retained for tens of *seconds*, suggesting the question of how time-varying neural activities maintain stable representations. Prior work shows that, if the neural dynamics are in the 'null space' of the representation - so that changes to neural activity do not affect the downstream read-out of stimulus information - then information can be retained for periods much longer than the time-scale of individual-neuronal activities. The prior work, however, requires precisely constructed synaptic connectivity matrices, without explaining how this would arise in a biological neural network. To identify mechanisms through which biological networks can self-organize to learn memory function, we derived biologically plausible synaptic plasticity rules that dynamically modify the connectivity matrix to enable information storing. Networks implementing this plasticity rule can successfully learn to form memory representations even if only 10% of the synapses are plastic, they are robust to synaptic noise, and they can represent information about multiple stimuli.

## 1 INTRODUCTION

Working memory is a key cognitive function, and it relies on us retaining representations of external stimuli even after they go away. Stimulus-specific elevated firing rates have been observed in the prefrontal cortex during the delay period of working memory tasks, and are the main neural correlates of working memory (Funahashi et al., 1993; Fuster & Alexander, 1971). Perturbations to the delay period neural activities cause changes in the animal's subsequent report of the remembered stimulus representation (Li et al., 2016; Wimmer et al., 2014). These elevated delay-period firing rates are not static but have time-varying dynamics with activities changing over timescales of tens of *milliseconds* (Brody et al., 2003a; Romo et al., 1999), yet information can be retained for tens of *seconds* (Fig. 1A). This suggests the question of how time-varying neural activities keep representing the same information.

Prior work from Druckmann & Chklovskii (2012) shows that, if the neural dynamics are in the "null space" of the representation - so that changes to neural activity do not affect the downstream read-out of stimulus information - then information can be retained for periods much longer than the time-scale of individual neuronal activities (called the FEVER model; Fig. 1B) . That model has a severe fine-tuning problem, discussed below. We identified a synaptic plasticity mechanism that overcomes this fine-tuning problem, enabling neural networks to learn to form stable representations.

While the dynamics of neurons in the FEVER model match that which is observed in the monkey prefrontal cortex during a working memory task (Murray et al., 2017), the model itself requires that the network connectivity matrix have one or more eigenvalues very near to unity. According to the Gershgorin Circle Theorem, this will almost surely not happen in randomly-connected networks: fine-tuned connectivity is needed. Druckmann & Chklovskii (2012) suggest a mechanism of Hebbian learning by which this connectivity can be learned. That mechanism requires the read-out weights to form a 'tight frame' , which will not necessarily be true in biological circuits. Thus, the prior work leaves it unknown how synaptic plasticity can form and/or maintain functional working memory networks. Here, we identify biologically plausible synaptic plasticity rules that can solve this fine-tuning problem without making strong assumptions like 'tight frame' representations. Our plasticity rules dynamically re-tune the connectivity matrix to enable persistent representations of

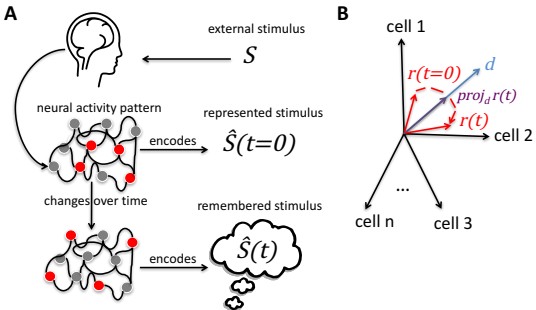

Figure 1: Stimulus representation in working memory. **(A)** When presented with an external stimulus, *s*, neural activity patterns initially encode an internal representation of that stimulus, *ŝ(t=0)*. These neural activity patterns change over time on timescales of tens of milliseconds, and yet somehow the same information is stored for up to tens of seconds. **(B)** While the firing rates, $r_i(t)$, change over time, information about stimulus, *ŝ(t)*, can be remain unchanged as long as the projection of the firing rates onto the "read-out" vector $\vec{d}$, remains constant (Druckmann & Chklovskii, 2012).

stimulus information. We specifically address parametric working memory, where the requirement is to remember continuous values describing several different variables or dimensions such as spatial locations or sound frequencies. For example, in the oculomotor delayed response task, subjects must remember the location of a target during a delay period (Funahashi et al., 1993). We perform experiments to demonstrate that networks using these plasticity rules are able to store information about multiple stimuli, work even if only a fraction of the synapses are tuned, and are robust to synaptic noise. We also show that these networks improve over time with multiple presented stimuli, and that the learning rules work within densely or sparsely connected networks.

## 2 MODEL

### 2.1 THE RATE-BASED NETWORK MODEL

We use a rate-based network like that of the FEVER model (Druckmann & Chklovskii, 2012) but with positive rectifying activation functions (ReLU - rectified linear units) (Carandini, 2004). We use standard linear dynamics in the network model:

$$\tau \frac{da_i}{dt} = -a_i(t) + \sum_j L_{ij} r_j(t),$$ 

(1)

where $a_i(t)$ is the internal state (membrane potential) of the $i^{th}$ neuron at time *t*, $r_j(t)$ is the output (firing rate) of the $j^{th}$ neuron, $\tau$ the time constant, and $L_{ij}$ represents the strength of synapse from neuron *j* to neuron *i*. Firing rates, $r_j(t)$, are given by a positive rectifying function of the internal states: $r_j(t) = [a_j(t)]_+$. After an external stimulus, *s*, is presented to the network, the network's representation of the stimulus, *ŝ(t)*, is obtained from a weighted combination of firing rates:

$$\hat{s}(t) = \sum_i d_i r_i(t),$$ 

(2)

where $d_i$ is the weight of the contribution of neuron *i* to the stimulus (the "read-out weight"). We first consider a single scalar stimulus value, and study the encoding of multiple stimuli in Sec. 3.5.

### 2.2 PLASTICITY RULES

To organize the network, we update the synaptic weights, $L_{ij}$, so as to minimize changes in the stimulus representation. To do this, we differentiated Equation 2 with respect to time, to calculate $\frac{d\hat{s}}{dt}$. We used gradient descent with respect to $L_{ij}$ on loss function $(\frac{d\hat{s}}{dt})^2$ to calculate the update rule:

$$\Delta L_{ij} = -\eta \frac{d\hat{s}}{dt} d_i \frac{dr_i}{da_i}(t) r_j(t),$$ 

(3)

where $\eta$ is the learning rate of the network and $\frac{dr_i}{da_i}(t)$ is the slope of the (ReLU) activation function. In section 3.6, we discuss the biological plausibility of this plasticity rule. We chose the elements of $\vec{d}$ to be positive based on candidate sources of the global error signal discussed in section 3.6. The source code and scripts for reproducing our experiments are available at [retracted for anonymity].

# 3 RESULTS

## 3.1 STIMULUS RETENTION IN THE SELF-ORGANIZING MEMORY NETWORK

To evaluate the performance of our working memory networks, we asked how well the networks could store information about a scalar stimulus value. We quantified the fraction of stimulus retained, $\frac{\hat{s}(t)}{\hat{s}(t=0)}$, over 3 seconds. This is the duration of heightened activity observed during working memory tasks (Funahashi et al., 1993). The networks were all-to-all connected (partial connectivity discussed in Sec. 3.3) and contained 100 neurons. We evaluated how well random networks without plastic synapses store information, by initializing each network with random neural activities, $\vec{a}(t=0)$, random read-out weights, $\vec{d}$, and random connection weight matrices, $L$, and simulating the dynamical evolution of the representation. We then compared these "constant random synapse" networks to ones with identical random initial conditions, but in which our plasticity rule (Eq. 3) dynamically updated the synapses (Figs. 2B, C). Finally, we compared both of these randomly-initialized networks to ones that had the fine-tuned connectivity specified by the FEVER model.

The stimulus value in the randomly-initialized plastic network initially decreased slightly, but the plasticity rules quickly reorganized the connectivity, and the representation remained constant after the first $\sim 50$ ms. In the networks with fixed random synaptic weights, the representation quickly decayed to 0 (Figs. 2A,B). Thus, our plasticity rule enables initially random networks to quickly become effective working memory systems.

To ensure that the success of our plasticity rule at forming an effective memory network was not limited to a fortuitous random initialization, we quantified the fraction of stimulus retained over 100 different networks, each with a different initial connectivity matrix, read-out vector, and initial activity vector $\vec{a}(t=0)$. In the FEVER networks, stimulus retention is perfect across all networks. The models with plastic random synapses perform almost as well as the FEVER models, but require some time to self-organize before the representations remain constant (Figs. 2B, C). In the random constant synapse networks, information is quickly lost (Fig. 2B).

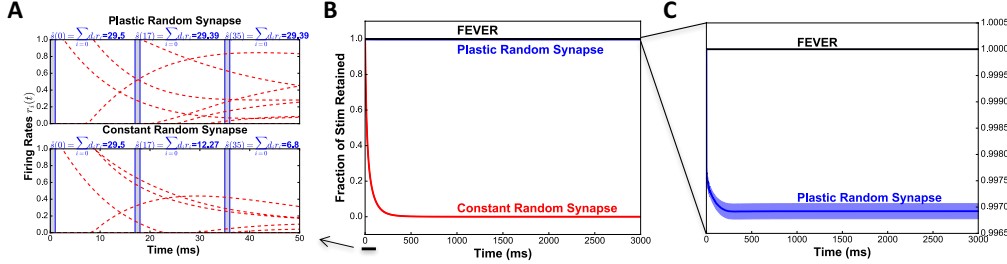

Figure 2: Stimulus retention in self-organizing memory networks. (**A**) Neural dynamics of the first 50 ms of a simulation of two different networks with the same random initial conditions. One network had constant random synapses (bottom panel), while the other one had plastic synapses that were updated via Eq. 3 (top panel). Red dashed lines show the firing rates of 10 of the neurons. The remembered stimulus values at times 0, 17 ms and 35 ms (indicated by the shaded vertical bars) are shown. In the plastic random network, the remembered stimulus value does not change much even though the neural activities keep evolving. (**B**) The average fraction of stimulus retained over 100 random initializations for the FEVER model (black), the plastic synapse model (blue) and the constant random synapse model (red). (**C**) A zoomed in look at the average fraction of stimulus retained for the FEVER model and the plastic random synapse model from (B). Shaded areas in B and C represent $\pm$ standard error of the mean.

### 3.2  NETWORK ROBUSTNESS

#### 3.2.1  NOISY SYNAPSES

Working memory must be robust to noise and imprecise components (Brody et al. (2003b)). We added Gaussian noise with mean 0 and standard deviation .00001 to demonstrate that FEVER networks with initially tuned constant synapses forgets the stimulus representation, whereas the network with randomly initialized plastic synapses is robust to the added noise (Fig. 3A). To determine how robust our networks are to noise, we simulated networks with various levels of noise added to the synaptic updates. We added Gaussian noise with mean 0 and standard deviation $\alpha$ times the update to the synapse, $\Delta L_{ij}$, where $\alpha$ was varied from 0 to 1. We quantified the fraction of stimulus retained for networks with various noise levels and found that the noise did not have a noticeable effect on the network performance (Fig. 3B). The fraction of stimulus retained is almost equivalent for all values of $\alpha$, with minor differences due to random initial conditions. This should not be surprising considering that multiplying error signals by random synaptic weights does not hinder learning, so long as the network is still being pushed down the loss gradient (Lillicrap et al., 2016).

#### 3.2.2  PARTIAL TUNING

While synapses are plastic, it is not known if *all* synapses change. To determine how well the network performs if only some synapses are updated, we simulated networks in which different fractions of the synapses were updated using Eq. 3: the other synapses were held constant. We quantified the fraction of stimulus retained by these networks (Figs. 4A, B). Even with just 10% of the synapses being tuned, the networks learn to store information about the stimulus. In hindsight, this makes sense. To store $n$ stimulus values, $n$ constraints must be satisfied by the connectivity matrix: it must have $n$ eigenvalues near 1 (we chose $n = 1$ for Figs. 4A, B). Because the connectivity matrix has many more than $n$ elements, many configurations can satisfy the constraint, so it is possible to satisfy the constraint without updating every synapse.

### 3.3  PARTIAL CONNECTIVITY

In the previous results, the connectivity was all-to-all (100%). Real neural circuits are not 100% connected. In visual cortex, for example, connection probabilities range from 50% to 80% for adjacent neurons (Hellwig, 2000). To ask if our synaptic update rule could self-organize partially connected networks, we simulated networks with different connection probabilities and with synapses updated using Eq. 3. We quantified the fraction of stimulus retained. Performance declines somewhat as connectivity decreases, but even networks with 10% connection probabilities can learn to store stimulus information (Fig. 4C). Experiments show that working memory performance declines with age, which correlates with a reduction in number of synapses (Peters et al., 2008).

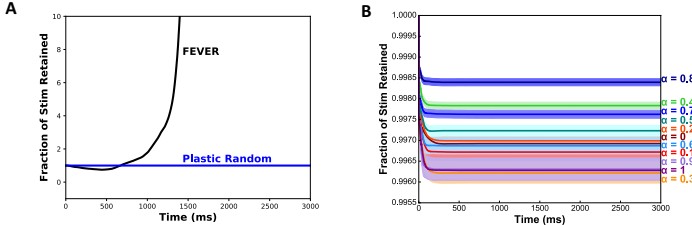

Figure 3: Network robustness to noise. **(A)** The fraction of stimulus retained for 100 random initializations for a FEVER network (black) and plastic random network (blue) with Gaussian noise mean 0 and standard deviation .00001. **(B)** The fraction of stimulus retained for 100 random initializations of networks with differing levels of synaptic update noise ($\alpha$). Shaded areas in A and B represent $\pm$ standard error of the mean.

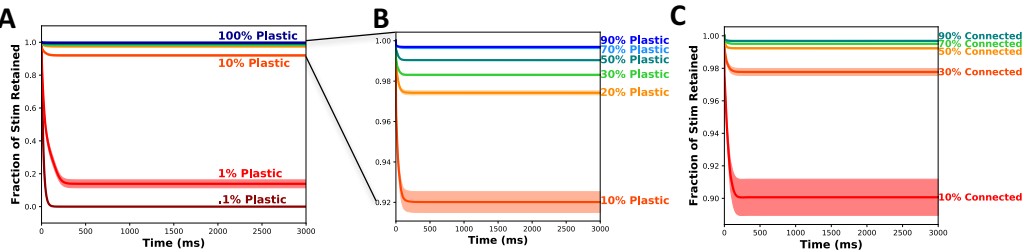

Figure 4: Network robustness to partial plasticity and partial connectivity. **(A)** The fraction of stimulus retained for 100 random initializations. Different lines are for different fractions of plastic synapses. **(B)** The fraction of stimulus retained over 100 random initializations. Different lines are for different connection probabilities. Shaded areas represent $\pm$ standard error of the mean.

## 3.4 Pre-training the Network

In the previous examples, each network is initialized with random connection weights. In reality, the working memory network will be continuously learning and will not start over with random connection weights when each new stimulus is presented. Consequently, we speculated that, once the network had learned to store one stimulus, it should be able to remember subsequently presented stimuli, even with minimal re-training. Relatedly, experimental work shows that performance in working memory tasks in children and young adults can be increased not only for trained tasks but for new tasks not part of the training: this coincides with strengthening of connectivity in the prefrontal cortex (Constantinidis & Klingberg, 2016).

To determine if our synaptic update rule enables the network to store new stimuli without further training, we first trained the networks (Eq. 3) to remember 1, 5 or 10 individual stimuli, one at a time: each new stimulus corresponded to another random initialization of the activity patterns $\vec{a}(t = 0)$. We quantified the networks' abilities to represent these training stimuli, and found that the networks performed better on each subsequent stimulus: training improved performance (Fig. 5A). Next, we asked if after training on 1, 5, or 10 prior stimuli, the network could store information about a new stimulus without any more synaptic updates. We found that a network was able to store information about a new stimulus after being trained on at least 1 previous stimulus (Fig. 5B). Once the connectivity weight matrix ($L$) has obtained one or more eigenvalues near unity, it is able to stably store novel stimuli without additional training.

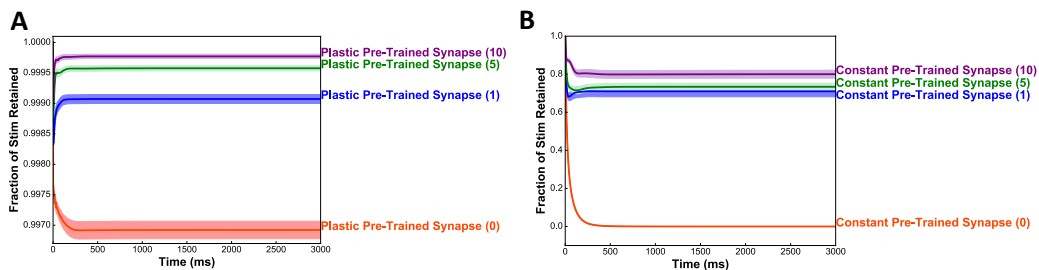

Figure 5: Training improves performance. **(A)** The fraction of stimulus retained over 100 random initializations for a plastic random synapse network that has seen 0 previous stimuli (orange), 1 previous stimulus (blue), 5 previous stimuli (green) or 10 previous stimuli (purple). **(B)** The fraction of stimulus retained over 100 random initializations for a constant synapse network that has previously been trained on 0, 1, 5, or 10 previous stimuli, but with no training during the simulation period shown. Color coding as in panel A. Shaded areas in A and B represent $\pm$ standard error of the mean.

## 3.5 REMEMBERING MULTIPLE STIMULI

The previous section shows how networks can self-organize to store information about one stimulus value, but working memory capacity in adult humans is typically 3–5 items (Cowan, 2010). To incorporate this working memory capacity into our models, we adapted the representation such that there were multiple read-out vectors, one for each stimulus value. We then derived plasticity rules via gradient descent on the squared and summed time derivatives of these representations: the loss was $\sum_k \left(\frac{d\hat{s}_k}{dt}\right)^2$. This led to the plasticity rule

$$\Delta L_{ij} = -\eta \sum_{k=1}^{n} \frac{d\hat{s}_k}{dt} d_{i_k} \frac{dr_i}{da_i}(t) r_j(t), \tag{4}$$

where $n$ is the number of stimuli to be remembered. We chose $n = 4$ for our experiments, and we quantified how well the networks remember these stimuli (S1-S4 in Fig. 6). (The networks can store up to 100 stimuli; data not shown). In our experiments, each neuron in the network contributed to the representation of every stimulus value: *in vivo*, most neurons are sensitive to multiple aspects of stimuli (Miller & Fusi, 2013). This is not a requirement: the models successfully represent multiple stimuli even when subsets of the neurons participate in each representation.

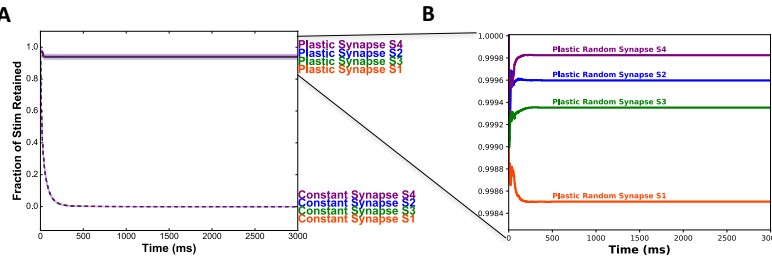

Figure 6: Remembering four things. **(A)** The average fraction of four stimuli retained for 100 randomly initialized networks with constant random synapses (dashed lines) and plastic random synapses (solid lines). Different colors are for different stimulus values. Shaded areas represent $\pm$ standard error of the mean. **(B)**. A zoomed in look at the average fraction of four stimuli retained in the plastic model in panel A. Shading omitted from lines in panel B for clarity.

## 3.6 BIOLOGICAL PLAUSIBILITY

Synaptic updates are thought to rely on synaptically local information, like the activities of the pre- and post-synaptic neurons. Our plasticity rules (Eqs. 3,4) involve this information, in addition to "global" error value(s) $\left\{\frac{d\hat{s}_k}{dt}\right\}$. Thus, we obtained three-factor rules: synaptic changes depend on the pre- and post-synaptic neurons' activities, and a global error signal (Lillicrap et al., 2016). There are two facets of the networks presented above that are unclear in their biological interpretations: the source (and precision) of the global error signal, and the symmetry of the feedback signals that convey the error information to the synapses. We discuss these issues in more detail below, and show that they are not firm requirements: our networks can learn to form memory representations even with more realistic asymmetric feedback, and with very coarse (even binary) error signals. Consequently, we demonstrate that synaptic plasticity mechanisms can successfully form self-organizing memory networks with minimal constraints.

### 3.6.1 SOURCES OF THE GLOBAL ERROR SIGNAL

Below, we propose two sources for the global error signal(s), $\left\{\frac{d\hat{s}_k}{dt}\right\}$: feedback to the neurons' apical dendrites (Fig. 7A), and neuromodulatory chemicals that modulate the plasticity (Fig. 7B). These are not mutually exclusive, and in either case, continuous training is not required for functioning working memory (Fig. 5): the feedback signals need not be constantly available. In both implementations, a downstream "read-out" system estimates $\hat{s}$, and sends that information (or information about its time derivative) back to the memory network. This means that, at first glance, the read-out weights $d_i$ must be accessed in two separate places: at the synapse, $L_{ij}$, to calculate the update,

and at the read-out layer to calculate the remembered stimulus value (Eq. 2). There are no known mechanisms that would allow this same $d_i$ value to be accessed at distantly-located regions of the brain (Lillicrap et al., 2016), posing a challenge to the biological plausibility of our networks. To address this issue, and show that known feedback and neuromodulatory mechanisms could implement our memory circuits, we later consider random (asymmetric) feedback weights, where the read-out weight used to estimate $\hat{s}$, differs from the one used to update the synaptic weights (Fig. 7C).

Calculating the Global Error Signal Locally Using Segregated Dendrites: The global error signal could be calculated locally by each neuron, by exploiting the fact that, in pyramidal cells, feedback arrives at the apical dendrites, and modulates synaptic plasticity at the basal dendrites (where information comes in from other cells in the memory network) (Lillicrap et al., 2016; Guergiuev et al., 2016) (Fig. 7A). Here, a readout layer provides feedback to the apical dendrites that specifies the represented stimulus value $\hat{s}(t)$: the weight of the feedback synapse to neuron $i$ from read-out neuron $k$ is $d_{i_k}$, and so the apical dendrite receives a signal $\sum_k d_{i_k}\hat{s}_k(t)$. The apical dendrites track the changes in these feedback signals, sending that information ($\sum_k d_{i_k}d\hat{s}_k/dt$) to basal dendrites via the soma. Correlating those signals with the pre- and post-synaptic activity at each of the synapses on the basal dendrites, the synaptic updates specified by Eq. 4 are obtained. Thus, the neurons locally compute the synaptic updates.

Signalling the Global Error Signal with Neuromodulators: Alternatively, the global error signal(s) could be communicated throughout the network by neuromodulators, like dopamine, acetylcholine, serotonin, or norepinephrine. These have all been shown to be important in synaptic plasticity in the prefrontal cortex and in working memory (Meunier et al., 2017). This is reward learning, with the reward values coming from the neuromodulator concentrations. Experimental work shows that synapses have activity-dependent "eligibility traces" that are converted into changes in synaptic strength by reward-linked neuromodulators (He et al., 2015). In this scenario the concentration of different modulators tracks the error signals, $\frac{d\hat{s}_k}{dt}$, and the densities of the receptors to the modulators at each synapse are $d_{i_k}$. Thus, at each synapse, the modulators bring information $\sum_k d_{i_k}d\hat{s}_k/dt$ that, when correlated with the pre- and post-synaptic activities, yields the updates from Eq. 4 (Fig. 7B).

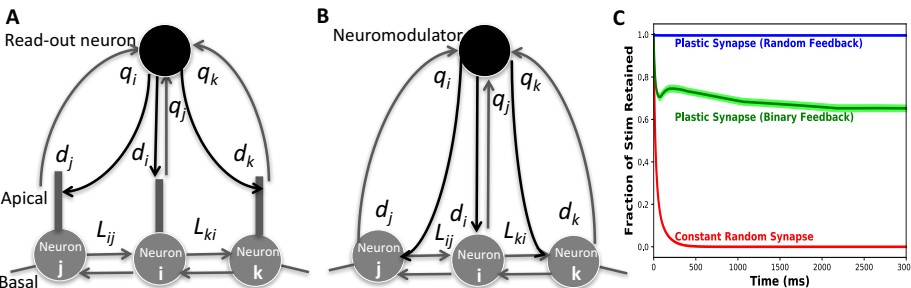

Figure 7: Potential biological implementations. **(A)** Cartoon depicting error signals being calculated locally by the neurons' apical dendrites (Guergiuev et al., 2016). The read-out neuron calculates the remembered stimulus value via Eq. 5, where $q_i$ is the weight of the synapses from cell $i$ in the memory circuit to the read-out neuron. The apical dendrite calculates the error signal, weighted by $d_i$, by subtracting feedback values at adjacent time points. The basal dendrite receives inputs from neurons in the working memory circuit. The soma transmits the error signal to the basal dendrites, where the synaptic updates, $L_{ij}$ are calculated by correlating the pre- and post-synaptic activities with the error signal. **(B)** Cartoon depicting error signals being communicated by neuromodulators. Neurons in the modulatory system calculate the remembered stimulus values via Eq. 5, where $q_i$ is the weight of the synapse from cell $i$ in the memory circuit to the read-out neuron. That cell releases an amount of neuromodulator that tracks the changes in the represented stimulus value. The modulatory chemical affects synaptic plasticity by an amount that depends on the receptor density, $d_i$, at the synapses. Both implementations work with asymmetric feedback: the weights to the read-out neuron from cell $i$ ($q_i$) will not necessarily match the weights, $d_i$, with which cell $i$ receives the feedback signals. **(C)** The fraction of stimulus retained over 100 random initializations of a network with plastic random synapses with random feedback and readout weights ($q \neq d$) (blue), plastic random synapses with random feedback and readout weights and binary error signals ($\frac{d\hat{s}_k}{dt} \in \{\pm 1\}$) (green), and constant synapses (red). The shaded areas represent $\pm$ standard error of the mean.

### 3.6.2 LEARNING WITH ASYMMETRIC RANDOM FEEDBACK WEIGHTS AND BINARY ERROR SIGNALS

The discussion above shows that it is critical to implement our memory networks with asymmetric feedback weights. To do this, we let the top-down feedback impinge on the neurons at synapses with weights $d_i$, and the read-out layer calculate the remembered stimulus as:

$$\hat{s}(t) = \sum_i q_i r_i(t), \tag{5}$$

where $d \neq q$ (Lillicrap et al., 2016). Here, the values of $q_i$ are randomly drawn, independently from $d_i$, and so we refer to this as asymmetric random feedback.

To test whether networks with this asymmetric feedback could learn to store stimulus information, we simulated such networks and quantified the fraction of stimulus retained. The results (Fig. 7C, upper curve) show that even with asymmetric feedback, networks can learn to store stimulus information. This makes sense because both $q$ and $d$ contain only positive elements, so the feedback update signal to each synapse has the same sign as the update calculated from gradient descent. Thus, the synaptic updates with asymmetric feedback are generally in the same direction as those obtained from gradient descent (i.e., the angle between the update vectors, and those from true gradient descent, is less than $90^o$), which suffices for learning (Lillicrap et al., 2016).

Next, we wondered whether our networks require precise error signals $\frac{d\hat{s}_k}{dt}$ to learn to form memory representations, or whether coarser feedback would suffice: if coarser signals suffice, this removes any fine-tuning requirement "hidden" in the precision of the feedback. To answer this question, we repeated the simulations with our asymmetric random feedback networks, but binarized the error signals used in the synaptic plasticity, via the sign function: $\text{sign}(\frac{d\hat{s}_k}{dt}) \in \{\pm 1\}$. The updates are now reduced to either Hebbian or anti-Hebbian learning rules, depending on the sign of the error signal. The results (Fig. 7C, lower curve) show that with asymmetric and binary feedback, the networks can still learn to form memory representations, albeit not quite as well as in the case of highly precise feedback signals (Fig. 7C, upper curve).

## 4 DISCUSSION

We derived biologically plausible synaptic plasticity rules through which networks self-organize to store information in working memory. Networks implementing these plasticity rules are robust to synaptic noise, to having only some of the synapses updated, and to partial connectivity. These networks can store multiple stimuli and have increased performance after previous training. We suggest two candidate sources for the global error signal necessary for the plasticity rule, and demonstrate that our networks can learn to store stimulus information while satisfying the added requirements imposed by these biological mechanisms. This flexibility suggests that other types of synaptic plasticity updates may also be able to organize working memory circuits.

The results presented here were obtained for networks of 100 neurons – as opposed to larger networks – to speed up the simulations. Tests on networks with 10,000 neurons show that the update rule works in larger networks. The optimal learning rate, $\eta$, decreases as the network size increases. Aside from network size, a potential caveat in using a rate-based network model is losing information about spike-timing dependency. A future direction would be to create a spike-based model and determine what, if anything, must be adjusted to account for spike timing, and for the discretization that spiking neurons entail for the information shared between cells.

Along with understanding how information is stored in working memory, this work may have implications in training recurrent neural networks (RNNs). Machine learning algorithms are generally unrealistic from a biological perspective: most rely on non-local synaptic updates or symmetric synapses. We show that recurrent networks can learn to store information using biologically plausible synaptic plasticity rules which require local information plus a global error signal (or signals), that can be calculated on the apical dendrite or via neuromodulators. This same setup could be utilized in RNNs to make them more biologically realistic. This would let us better understand how the brain learns, and could lead to novel biomimetic technologies: prior work on biologically realistic machine learning algorithms has led to hardware devices that use on-chip learning (Knag et al., 2015; Zylberberg et al., 2011). Synaptically local updates do not have to be coordinated over all parts of the chip, enabling simpler and more efficient hardware implementations.

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
