# OpenReview forum: "A Self-Organizing Memory Network"
_ICLR.cc/2018/Conference — Reject_

### Official Review · AnonReviewer3 · 2017-11-25
**Too many questions to make sense of this work**

**Rating:** 3
**Confidence:** 2

**Review:**

This is a great discussion on an interesting problem in computational neuroscience, that of holding an attractor memory stable even though individual neurons fluctuate. The previously published idea is that this is possible when the sum of all these memory neurons remain constant for the specific readout network, which is possible with the right dependency of the memory neurons. While this previous work relied on fine tuned weights to find such solutions, this work apparently shows that a gradient-based learning rule can finds easily robust solutions.

Unfortunately, I am a bit puzzled about this paper in several ways. To start with I am not completely sure why this paper is submitted to ICLR. While it seems to address a technical issue in computational neuroscience, the discussion on the impact on machine learning is rather limited. Maybe some more discussion on that would be good.

My biggest concern is that I do not understand the model as presented. A recurrent network of the form in eq 1 is well known as an attractor network with symmetric weights. However, with the proposed learning rate this might be different as I guess this is not even symmetric. Shouldn’t dr_i/da_i = 1 for positive rates as a rectified linear function is used? I guess the derivation of the learning rule (eq.3) is not really clear to me. Does this not require the assumption of stationary single recurrent neuron activities to get  a_i=\sum L_ij r_j? How do the recurrent neurons fluctuate in the first experiments before noise is introduced? I see that the values change at the beginning as shown in Fig1a, but do they continue to evolve or do the asymptotic? I think a more careful presentation of this important part of the paper would be useful.

Also, I am puzzled about the readout, specifically when it comes to multiple memories. It seems that separate memories have different readout nodes as the readout weights has an additional index k in this case. I think the challenge should be to have multiple stable states that can be read out in the same pathway. I might miss here something, though I think that without a more clear explanation the paper is not clear for a novel reader.

And how about Figure 3a? Why is the fever model suddenly shooting up? This looks rather than a numerical error.
In summary, while I might be missing some points here, I can not make sense of this paper at this point.

---

### Official Review · AnonReviewer2 · 2017-11-27
**The proposed model shows good performance but it is rather speculative, the paper lacks more in-depth analysis, and the relevance for the conference is unclear**

**Rating:** 4
**Confidence:** 4

**Review:**

A neural network model consisting of recurrently connected neurons and one or more readouts is introduced which aims to retain some output over time. A plasticity rule for this goal is derived. Experiments show the robustness of the network with respect to noisy weight updates, number of non-plastic connections, and sparse connectivity. Multiple consecutive runs increase the performance; furthermore, remembering multiple stimuli is possible. Finally, ideas for the biological implementation of the rule are suggested.

While the presentation is generally comprehensible a number of errors and deficits exist (see below). In general, this paper addresses a question that seems only relevant from a neuroscience perspective. Therefore, I wonder whether it is relevant in terms of the scope of this conference. I also think that the model is rather speculative. The authors argue that the resulting learning rule is biologically plausible. But even if this is the case, it does not imply that it is implemented in neuronal circuits in the brain. As far as I can see, there exists no experimental evidence for this rule.

The paper shows the superiority of the proposed model over the approach of Druckmann & Chkolvskii (2012), however, it lacks in-depth analysis of the network behavior. Specifically, it is not clear how the information is stored. Do neurons show time-varying responses as in Druckmann & Chkolvskii (2012) or do all neuron stabilize within the first 50 ms (as in Fig. 2A, it is not detailed how the neurons shown there have been selected)? Do the weights change continuously within the delay period or do they also converge rapidly? This question is particularily important when considering multiple consecutive trials (cf. Fig. 5) as it seem that a specific but constant network architecture can retain the desired stimulus without further plasticity. Weight histograms should be presented for the different cases and network states. Also, since autapses are allowed, an analysis of their role should be performed. This information is vital to compare the model to findings from neuroscience and judge the biologic realism.

The target used is \hat{s}(t) / \hat{s}(t = 0), this is dubbed "fraction of stimulus retained". In most plots, the values for this measure are <= 1, but in Fig. 3A, the value (for the FEVER network) is > 1. Thus, the name is arguably not well-chosen: how can a fraction of remembrance be greater than one? Also, in a realistic environment, it is not clear that the neuronal activities decay to zero (resulting in \hat{s}(t) also approaching zero). A squared distances measure should therefore be considered.

It is not clear from the paper when and how often weight updates are performed. Therefore, the biologic plausability cannot be assessed, since the learning rule might lead to much more rapid changes of weights than the known learning rules in biological neural networks. Since the goal seems to be biologic realism, generally, spiking neurons should be used for the model. This is important as spiking neural networks are much more fragile than artificial ones in terms of stability.

Further remarks:

- In Sec. 3.2.1, noise is added to weight updates. The absolute values of alpha are hard to interpret since it is not clear in what range the weights, activities, and weight updates typically lie.

- In Sec. 3.2.2 it is shown that 10% plastic synapses is enough for reasonable performance. In this case, it should be investigated whether the full network is essential for the memory task at all (especially since later, it is argued that 100 neurons can store up to 100 stimuli).

- For biologic realism, just assuming that the readout value at t = 0 is the target seems a bit too simple. How does this output arise in the first place? At least, an argument for this choice should be presented.


Remarks on writing:

- Fig. 1A is too small to read.

- The caption of Fig. 4C is missing.

- In Fig. 7AB, q_i and q_j are swapped. Also, it is unclear in the figure to which connection the ds and qs belong.

- In 3.6.1, Fig. 7 is referenced, but in the figure the terminology of Eq. 5 is used, which is only introduced in Sec. 3.6.2. This is confusing.

- The beginning of Sec. 3.6 claims that all information is local except d\hat{s}_k / dt, but this is not the case as d_i is not local (which is explained later).

- The details of the "stimulus presentation" (i.e. it is not performed explicitly) should be emphasised in 2.1. Also, the description of the target \hat{s} is much clearer in 3.4 than in 2.1 (where it should primarily be explained).

- The title of the citation Cowan (2010) is missing.

- In Fig. 2A, the formulas are too small too read in a printed version.

- In Sec. 3.6.1 some sums are given over k, but k is also the index of a neuron in Fig. 7A (which is referenced there), this can be ambiguous and could be changed.

---

### Official Review · AnonReviewer4 · 2017-12-16
**Learning to remember**

**Rating:** 4
**Confidence:** 4

**Review:**

This paper presents a self-organizing (i.e. learned) memory mechanism in a neural model.   The model is not so much an explicit mechanism, rather the paper introduces an objective function that minimizes changes in the signal to be memorized.

The model builds on the FEVER model (Druckmann and Chklowskii, 2012) and stays fairly close to the framework and goals laid out in this paper. The contribution offered in this paper is a gradient-based weight update (corresponding to the objective function being the square of the temporal derivative of the signal to memorize). This naturally extends the FEVER framework to non-linear dynamics and allows the model to learn to be more robust to weight noise than the original FEVER model.

The paper goes on to show a few properties of the new memory model including it's ability to remember multiple stimuli and sensitivity to various degrees of connectivity and plasticity. The authors also demonstrate that the update rule can be implemented with a certain degree of biological plausibility. In particular, they show that the need for the same weights to be used in the forward propagation of activity and the backward propagation of gradient can be relaxed. This result is consistent with similar findings in the deep learning literature (Lillicrap et al., 2016).

For the reader interested in models of working memory and how it can be implemented in dynamic neural hardware, I believe this paper does contribute something interesting to this field of study.

I have two principle concerns with the paper. First, it seems that ICLR is not the right venue for this work. While ICLR has certainly published work with a strong neuro-scientific orientation, I believe this paper offers relatively little of interest to those that are not really invested in the models under consideration. The task that is considered is trivial - maintain a constant projection of the activity of the network activations. I would expect the model to attempt to do this in the service of another, more clearly useful task. In the RNN literature, there exists tasks such as the copy task that test memory, this model is really at a layer below this level of task sophistication.

Second, while the set of experiments are fair and appropriate, they also seem quite superficial. There is a lack of analysis of the consequences of the learning objective on the network dynamics. There just does not seem to be the same level of contribution as I would expect from an ICLR paper.

---

### Decision · Program_Chairs · 2018-01-29
**ICLR 2018 Conference Acceptance Decision**

**Decision:**

Reject

**Comment:**

This work extends Druckmann and Chklowskii, 2012 and demonstrates some interesting properties of the new model. This would be of interest to a neuroscience audience, but the focus is off for ICLR.